# Routes of Albumin Overload Toxicity in Renal Tubular Epithelial Cells

**DOI:** 10.3390/ijms24119640

**Published:** 2023-06-01

**Authors:** Theodoros Eleftheriadis, Georgios Pissas, Spyridon Golfinopoulos, Maria Efthymiadi, Christina Poulianiti, Maria Anna Polyzou Konsta, Vassilios Liakopoulos, Ioannis Stefanidis

**Affiliations:** Department of Nephrology, Faculty of Medicine, University of Thessaly, Biopolis, Mezourlo Hill, 41110 Larissa, Greece; gpissas@msn.com (G.P.); spygolfin@yahoo.gr (S.G.); mariaeuthimiadi@hotmail.com (M.E.); christinepoulianiti@yahoo.gr (C.P.); polyzoukonsta@yahoo.com (M.A.P.K.); liakopul@otenet.gr (V.L.); stefanid@uth.gr (I.S.)

**Keywords:** albumin, renal tubular epithelial cells, endoplasmic reticulum stress, DNA damage, apoptosis, senescence, epithelial-to-mesenchymal transition

## Abstract

Besides being a marker of kidney disease severity, albuminuria exerts a toxic effect on renal proximal tubular epithelial cells (RPTECs). We evaluated whether an unfolded protein response (UPR) or DNA damage response (DDR) is elicited in RPTECs exposed to high albumin concentration. The deleterious outcomes of the above pathways, apoptosis, senescence, or epithelial-to-mesenchymal transition (EMT) were evaluated. Albumin caused reactive oxygen species (ROS) overproduction and protein modification, and a UPR assessed the level of crucial molecules involved in this pathway. ROS also induced a DDR evaluated by critical molecules involved in this pathway. Apoptosis ensued through the extrinsic pathway. Senescence also occurred, and the RPTECs acquired a senescence-associated secretory phenotype since they overproduced IL-1β and TGF-β1. The latter may contribute to the observed EMT. Agents against endoplasmic reticulum stress (ERS) only partially alleviated the above changes, while the inhibition of ROS upregulation prevented both UPR and DDR and all the subsequent harmful effects. Briefly, albumin overload causes cellular apoptosis, senescence, and EMT in RPTECs by triggering UPR and DDR. Promising anti-ERS factors are beneficial but cannot eliminate the albumin-induced deleterious effects because DDR also occurs. Factors that suppress ROS overproduction may be more effective since they could halt UPR and DDR.

## 1. Introduction

Chronic kidney disease (CKD) affects more than 10% of the population worldwide and is one of the leading causes of death [1]. Consequently, there is an urgent need to apply the existent and discover new preventive and therapeutic measures. Typically, the glomerular filtration rate was used for CKD definition and classification. Still, since 2011, an albuminuria stage was added as the presence and quantity of albuminuria can predict the risk for CKD progression to end-stage renal disease and mortality [2].

Normally, a relatively small quantity of albumin passes the glomerular filtration barrier. Most of it is reabsorbed by the renal proximal tubular epithelial cells (RPTECs) and either catabolized or transcytosed. Thus, albuminuria occurs in kidney diseases that affect the proximal tubule, and, more frequently, the glomeruli [3,4].

Blockers of the renin–angiotensin system (RAS), and, more recently, sodium-glucose co-transporter-2 (SGLT-2) inhibitors are used for decreasing CKD progression. These agents reduce intraglomerular pressure, which may have a direct beneficial effect on the glomeruli, but, by doing so, they also reduce albuminuria [5,6,7]. The latter may play a significant role in the beneficial impact of RAS blockers and SGLT-2 inhibitors. Besides being a kidney, especially glomerular, disease severity marker, albuminuria is toxic to the renal tubules, contributing to CKD progression.

Regarding the mechanisms involved in albumin toxicity in RPTECs, most experimental studies suggest that albumin overload elicits an unfolded protein response (UPR) and endoplasmic reticulum stress (ERS) in RPTECs [8,9,10,11]. This is interesting since various anti-ERS agents are under development, and some of them have already been approved for clinical use, albeit in clinical entities other than CKD [12,13]. On the other hand, a recent study showed that albumin overload induces DNA damage in RPTECs, and the subsequent DNA damage response (DDR) results in cellular senescence [14].

Since albuminuria is a significant prognosticator of renal outcome in patients with CKD [2], a clarification of the exact mechanisms involved in albumin-induced toxicity in RPTECs is of particular interest as it may identify specific targets for new therapeutic agents. For this purpose, in the current study, we evaluated in RPTEC cultures whether albumin induces a UPR or a DDR, as well as the downstream pathways that may lead to the deleterious outcomes of apoptosis, senescence, or epithelial-to-mesenchymal transition (EMT) [15,16]. With the introduction of anti-ERS agents into clinical practice [12,13], it would be beneficial to elucidate the significance of this pathway, as well as the role of DDR, in the toxicity of albumin overload in RPTECs. This clarification would provide a more accurate understanding of the potential impact of anti-ERS agents on patients with albuminuria.

RPTEC apoptosis contributes to CKD progression obviously by decreasing the functional renal tissue [17]. Cellular senescence is a state of permanent cell cycle arrest. A large proportion of senescent RPTECs precludes the regeneration of apoptotic RPTECs, contributing to the progression of CKD. In addition, senescent cells acquire a senescence-associated secretory phenotype and produce various proinflammatory and profibrotic cytokines, aggravating renal damage [18]. The profibrotic cytokines may act in adjacent cells and transform them into myofibroblasts inducing tubulointerstitial fibrosis, in which all long-lasting kidney diseases culminate [19]. In the case of RPTECs, EMT may take place [20]. Interestingly, even in various glomerular diseases, the degree of tubulointerstitial fibrosis is the best prognosticator of renal outcome [19,21].

## 2. Results

### 2.1. Albumin Overload Triggers ROS Production and UPR

Lactate dehydrogenase (LDH) release assay showed that albumin overload does not induce cell necrosis. In addition, the anti-ERS agents tauroursodeoxycholic acid (TUDCA) and 4-Phenylbutyric acid (4-PBA) were not cytotoxic at the used concentrations (Figure 1A).

High albumin concentration induced a reactive oxygen species (ROS) burst in RPTECs. Neither TUDCA nor 4-PBA affected albumin overload-induced ROS production (Figure 1B). As a result of oxidative stress, protein modification occurred as the level of 4-Hydroxynonenal (4-HNE)-modified proteins increased. The anti-ERS agents partially decreased protein modification (Figure 1C,D).

Protein modification in RPTECs exposed to a high albumin concentration triggered a UPR since the phosphorylated PKR-like ER kinase (p-PERK) level increased (Figure 1C,E), the level of its substrate phosphorylated eukaryotic translation initiation factor-2α (p-eIF2α) was enhanced (Figure 1C,G), and the activating transcription factor-3 (ATF3) level was upregulated (Figure 1C,I). TUDCA and 4-PBA were ameliorated but did not eliminate all the above changes.

### 2.2. Albumin Overload Induces a DDR

Albumin overload caused DNA damage, as detected by the enhanced phosphorylated histone H2AX (γ-H2AX) level (Figure 2A,B). TUDCA and 4-PBA decreased, to some extent, the DNA damage. DNA damage in RPTECs exposed to high albumin concentration elicited a DDR since the phosphorylated ataxia telangiectasia mutated kinase (p-ATM) level increased (Figure 2A,C), resulting in the phosphorylation of its substrate p53 (Figure 2A,E), and eventually in the upregulation of the total p53 (Figure 2A,F). TUDCA and 4-PBA reduced to some extent all of the above albumin overload-induced changes.

### 2.3. Albumin Overload Triggers the Extrinsic but Not the Intrinsic Apoptotic Pathway

Albumin overload left the intrinsic apoptotic pathway unaffected, with the C/EBP homologous protein (CHOP) (Figure 3A,B), Bcl-2-associated X protein (Bax) (Figure 3A,C), and cleaved caspase-9 (CC-9) (Figure 3A,D) levels remaining relatively stable. TUDCA and 4-PBA did not change the levels of the above proteins.

On the contrary, the exposure of the RPTECs to high albumin concentration triggered the extrinsic apoptotic pathway, as death receptor-5 (DR5) (Figure 3A,E) and cleaved caspase-8 (CC-8) (Figure 3A,F) were significantly upregulated. Eventually, apoptosis ensued since the level of cleaved caspase-3 (CC-3) increased (Figure 3A,G). TUDCA and 4-PBA ameliorated, but did not eliminate, the above changes. 

### 2.4. Albumin Overload Induces Cellular Senescence and EMT

Albumin overload increased the expression of the cell cycle arrest inducers p21 (Figure 4A,B) and p16 (Figure 4A,C). In addition, it decreased the level of the cell proliferation marker Ki-67 (Figure 4A,D) and enhanced the cellular senescence marker β-galactosidase (GLB-1) (Figure 4A,E). TUDCA and 4-PBA ameliorated but did not eliminate all the aforementioned albumin-induced changes.

RPTECs exposed to high albumin concentration acquired a senescence-associated secretory phenotype since they overproduced interleukin-1β (IL-1β) (Figure 4F) and transforming growth factor-β1 (TGF-β1) (Figure 4G). TUDCA and 4-PBA significantly decreased the albumin-induced IL-1β and TGF-β1 overproduction.

The exposure of RPTECs to high albumin concentration resulted in EMT as the level of α-smooth muscle actin (α-SMA) increased (Figure 4A,H). TUDCA and 4-PBA ameliorated the albumin-induced α-SMA upregulation.

### 2.5. Albumin Overload Triggers UPR and DDR and Induces Apoptosis, Senescence, or EMT by Increasing ROS

To evaluate whether the inhibition of both UPR and DDR eliminates the detrimental effects of high albumin concentration in RPTECs, we used N-acetylcysteine (NAC) to suppress ROS overproduction, which triggers UPR and DDR. LDH release assay revealed that at the used concentration, NAC was not cytotoxic for the RPTECs (Figure 5A). NAC completely abolished the rise of ROS levels in RPTECs exposed to high albumin concentration (Figure 5B).

NAC prevented UPR and DDR induced by the high albumin concentration since it normalized the albumin-induced increase in the p-PERK (Figure 5C,D) and γ-H2AX levels (Figure 5C,F), respectively.

By preventing UPR and DDR, NAC prevents the harmful effects of albumin overload in RPTECs, apoptosis, senescence, and EMT. NAC normalized the albumin-induced upregulation of CC3 (Figure 5C,G), GLB-1 (Figure 5C,H), and α-SMA levels (Figure 5C,I).

## 3. Discussion

Albuminuria frequently accompanies CKD. The presence, and the greater the degree of albuminuria, the worse the renal outcome [2]. Besides being a marker of the severity of kidney, particularly glomerular, disease, albuminuria exerts a direct toxic effect on RPTECs. The latter is significant because almost all long-lasting kidney diseases culminate in tubulointerstitial fibrosis [19]. The degree of the latter is the best pathological marker for predicting CKD progression to ESRD [19,21].

In this study, we evaluated the routes of albumin overload toxicity in RPTECs. Using a cell culture system, we showed that albumin overload induces ROS production in RPTECs. It is known that albumin overload causes ROS production, but the exact mechanism is not well established [9,22]. Generally, lysosomal overload induces ROS production with an unclarified mechanism [23]. Since most studies attribute albumin toxicity in RPTECs to ERS [8,9,10,11], we used two well-established ERS inhibitors, TUDCA and 4-PBA [15]. Both compounds inhibit ERS due to their chaperoning activity [24,25], and experimental studies have confirmed that they protect RPTECs from ERS [26,27,28,29]. The fact that these substances are already in clinical use, 4-PBA to treat congenital diseases in the urea cycle [30], and TUDCA for treating primary biliary cholangitis [31], makes them particularly appealing for further investigation since we already know much about their pharmacologic properties and side effects. Currently, both medications are under investigation as anti-ERS agents for the treatment of various diseases [24,30], and recently a combination of them has been approved for the treatment of amyotrophic lateral sclerosis [13]. LDH release assay indicated that these anti-ERS substances are not toxic at the concentrations used in our experiments. Another interesting result from the LDH release assay was the absence of cell necrosis in RPTECs exposed to high albumin concentrations. On the contrary, other conditions that trigger ROS production, such as anoxia-reoxygenation, are known to induce ferroptotic cell necrosis [32]. Neither TUDCA nor 4-PBA affected ROS production. 

In RPTECs exposed to albumin, oxidative stress-induced protein modification was assessed by the level of 4-HNE-modified proteins. Both TUDCA and 4-PBA decreased to a great extent the level of 4-HNE-modified proteins. Protein modification may alter protein conformation resulting in UPR. Indeed, albumin overload induces a UPR assessed by the level of PERK phosphorylation, an early event in the UPR cascade. Activated p-PERK phosphorylates eIF2a, altering the translational program of the cell. The above results in the upregulation of the transcription factor ATF3 [12,15,33], which has also been detected in our experiments. TUDCA and 4-PBA ameliorated the p-PERK, p-eIF2a, and ATF3 upregulation.

Although most studies focus on albumin-induced RPTEC toxicity through the UPR pathway, at least one suggests that a DDR is elicited and may lead to cellular senescence [14]. Indeed, our experiments showed that albumin-induced ROS overproduction causes DNA damage, assessed by the level of γ-H2AX. Subsequently, DDR starts as ATM is phosphorylated. The latter activates ATM, which phosphorylates p53 [34]. Phosphorylated p53 dissociates with E3 ubiquitin-protein ligase Mouse double minute 2 homolog (Mdm2), which protects p53 from degradation by the proteasome, increasing its level [35]. Both TUDCA and 4-PBA lessen, to some extent, the above changes, possibly due to the known interplay between UPR and DDR [36]. However, an additional, unestablished pharmacological action of these compounds cannot be excluded. 

Apoptosis is a possible deleterious outcome of RPTEC exposure to high albumin concentration. Our results showed that albumin overload induces apoptosis in RPTECs assessed by the level of activated cleaved caspase-3 in which all the apoptotic pathways converge [37]. Then we evaluated which of the two apoptotic pathways, the intrinsic or the extrinsic, are triggered by the high albumin concentration. Surprisingly, although albumin overload led to p53 increase—which induces the transcription of many proapoptotic genes, including Bax [35], and UPR upregulates CHOP, which does the same [38,39]—we did not find increased levels of CHOP and Bax in RPTECs exposed to albumin. In addition, the level of activated cleaved caspase-9, the signature caspase of the intrinsic apoptotic pathway, remained unaffected [37]. On the contrary, we found that albumin overload triggers the extrinsic apoptotic path since the level of activated cleaved caspase-8 was upregulated [37]. The latter is likely the result of DR5 upregulation, a known target of the p53 transcription factor [40]. As noted, albumin increased p53 through the DDR [35,41]. UPR can also upregulate p53 since ATF3 interacts with p53 preventing its degradation [41,42]. The latter may explain the observed beneficial effect of TUDCA and 4-PBA in ameliorating, to some extent, apoptosis.

Another deleterious outcome that cells can encounter under stress conditions is senescence. Senescent cells enter a permanent cell cycle arrest condition, and, in addition, acquire a senescence-associated secretory phenotype, secreting proinflammatory and profibrotic cytokines and affecting the adjustment cells. In the case of CKD, senescent cells cannot dedifferentiate and proliferate to replace apoptotic cells. In addition, they secrete proinflammatory cytokines, aggravating inflammation and accelerating kidney injury. Finally, the profibrotic cytokines accelerate tubulointerstitial fibrosis, in which most kidney diseases culminate [18,43]. In our experiments, albumin overload caused RPTEC senescence. It increased p21 and p16, critical cell cycle arrest factors, and senescence markers. The upregulation of p21 can be attributed to the p53 increase since the latter transcribes the p21 gene [35,44]. The rise of p16 is the result of ATF3 upregulation [45,46]. In addition, oxidative stress upregulates p16 through a p38 stress-activated protein kinase-dependent pathway [47]. High albumin concentration reduced RPTEC proliferation, assessed by the cell proliferation marker Ki-67 [48,49], and increased the typical cellular senescence marker GLB-1 [48,50]. In addition, RPTECs acquired a senescence-associated secretory phenotype as they overproduced IL-1β and TGF-β1. TUDCA and 4-PBA ameliorated, but did not eliminate, the above albumin-induced changes.

As noted, RPTECs exposed to high albumin concentrations overproduce TGF-β1, the archetypical profibrotic cytokine. It is likely that TGF-β1 upregulation, acting in an autocrine manner, induced EMT, assessed by the α-SMA expression in RPTECs [20]. Thus, albumin overload by inducing EMT favors renal fibrosis. Certainly, the same profibrotic cytokine acting in a paracrine manner may favor fibrosis further by promoting the transformation of regional fibroblasts to myofibroblasts or the mesenchymal transition of endothelial cells or pericytes [20,51]. Again, TUDCA and 4-PBA reduced, but did not eliminate, the albumin-induced EMT.

To confirm whether albumin-induced UPR and DDR are the major pathways that induce cellular apoptosis, senescence, and EMT in RPTECs, we repeated the experiments using the potent antioxidant NAC [52]. NAC was used in a high but not toxic concentration, as portrayed by the LDH release assay. Indeed, in RPTECs exposed to high albumin concentration, NAC normalized ROS levels and prevented UPR and DDR, assessed by p-PERK and γ-H2AX levels, respectively. Consequently, NAC protected RPTECs from albumin-induced apoptosis, evaluated by cleaved caspase-3, senescence assessed by GLB-1, and EMT evaluated by α-SMA. 

A limitation of our study lies in its in vitro nature. However, the strict conditions of our experimental system allowed us to evaluate the effects of high albumin concentration in isolated RPTECs, excluding other confounding factors. For instance, hemodynamic changes, inflammation, or high glucose concentration that characterize various CKD experimental models might trigger similar pathways to those induced by high albumin concentration, making the study of the latter problematic. Certainly, the simultaneous assessment of markers associated with DDR, ERS, apoptosis, senescence, and EMT in clinical samples obtained from patients with albuminuria would be highly intriguing.

Our results are depicted in Figure 6 and support that albumin overload causes cellular apoptosis, senescence, and EMT in RPTECs by triggering both UPR and DDR. Promising anti-ERS factors, such as TUDCA and 4-PBA, may be beneficial but cannot eliminate the albumin-induced deleterious effects because DDR also occurs. Research on factors that suppress ROS overproduction due to exposure to high albumin concentration may result in finding more effective agents that are capable of halting both UPR and DDR.

## 4. Materials and Methods

### 4.1. Cell Culture Conditions

Primary human RPTCEs (ScienCell, Carlsbad, CA, USA) were cultured in a Complete Epithelial Cell Medium/w kit, supplemented with epithelial cell growth supplement (epithelial growth factor, insulin, transferrin, L-glutamine, selenium, fetal bovine serum, and antibiotics) (cat. no. M6621; Cell Biologics, Inc., Chicago, IL, USA). The cells above were differentiated, well-characterized, passage one RPTECs. Cells were expanded in 75 cm^2^ flasks, and passage three cells were used for the experiments.

Cells were exposed or not to 30 mg/mL of bovine serum albumin (Rockland Immunochemicals, Pottstown, PA, USA). Tauroursodeoxycholic acid (TUDCA, cat. no. S3654; Selleck Chemicals, Munich, Germany) or 4-Phenylbutyric acid (4-PBA, cat. no. S3592; Selleck Chemicals) was used for suppressing ERS at a concentration of 2 mM. N-acetylcysteine (NAC, cat. no. S1623; Selleck Chemicals) was used as an antioxidant at a 3 mg/mL concentration. The BSA, 4-PBA, TUDCA, and NAC concentrations were selected after preliminary experiments with concentrations within the range used in previous studies [10,26,29,53,54,55]. RPTECs were cultured with 10 or 30 mg/mL BSA, and p-PERK was used as the outcome. We selected the latter concentration, which had the highest effect. For 4-PBA and TUDCA, RPTECs cultured with 30 mg/mL BSA were exposed 1 or 2 mM of these compounds. Again p-PERK was the outcome. We selected the concentration with the highest effect. We used 1 or 3 mg/mL of NAC, and ROS production was the outcome. Again, we selected the concentration with the highest effect. The LDH release assay revealed that the selected concentrations were not cytotoxic for the RPTECs.

RPTECs were cultured in 96-well plates (1 × 10^4^ cells) or 6-well plates (3 × 10^5^ cells) at 37 °C for 24 h in a humidified atmosphere containing 5% CO_2_. All experiments were performed in triplicates.

### 4.2. Detection of Proteins Involved in UPR and DDR Pathways

Specific cellular proteins were assessed in RPTECs cultured in 6-well plates. Cells were lysed with the T-PER tissue protein extraction reagent (Thermo Fisher Scientific Inc., Waltham, MA, USA), supplemented with protease and phosphatase inhibitors (Sigma-Aldrich; Merck Millipore, Darmstadt, Germany and Roche Diagnostics, Indianapolis, IN, USA, respectively). Total cellular protein concentration was measured with Bradford assay (Sigma-Aldrich; Merck Millipore), and 10 μg of protein from each sample was electrophoresed in a sodium dodecyl sulfate-polyacrylamide gel (4–12% Bis-Tris gels, Thermo Fisher Scientific Inc.) and then transferred onto a polyvinylidene fluoride (PVDF) membrane (Thermo Fisher Scientific Inc.). Western blot bands were visualized with the LumiSensor Plus Chemiluminescent HRP Substrate kit (GenScript Corporation, Piscataway, NJ, USA). PVDF membrane re-probing was performed with the Restore Western Blot Stripping Buffer (Thermo Fisher Scientific Inc.). The Image J software version 1.53f (National Institute of Health, Bethesda, MD, USA) was used for Western blot bands analysis. These experiments were performed in triplicates.

Primary antibodies were applied for 16 h at 4 °C, while secondary antibodies were applied for 30 min at room temperature. Primary antibodies were specific for the following proteins: 4-Hydroxynonenal modified proteins (4-HNE, 1:500, cat. no. ab46545 Abcam, Cambridge, UK), PKR-like ER kinase (PERK, 1:100, sc-sc-377400, Santa Cruz Biotechnology, Dallas, TX, USA), phosphorylated at Ser713 PERK (p-PERK, 1:1000, cat. no. 649401, BioLegend, San Diego, CA, USA), eukaryotic translation initiation factor-2α (eIF2α, 1:100, cat. no. sc-133132, Santa Cruz Biotechnology, Dalas, TX, USA), phosphorylated at Ser51 eIF2α (p-eIF2α, 1:1000, cat. no. 9721, Cell Signaling Technology, Danvers, MA, USA), activating transcription factor-3 (ATF-3, 1:500, cat. no. CSB-PA020022, Cusabio Technology, Wuhan, China), phosphorylated at Ser139 histone H2AX (γ-H2AX, 1:500, cat. no. NB100-2280, Novus Biologicals, Abingdon, Oxon, UK), ataxia telangiectasia mutated kinase (ATM, 1:1000, cat. no. 2873, Cell Signaling Technology), phosphorylated at Ser1981 ATM (p-ATM, 1:1000, cat. no. 5883, Cell Signaling Technology); tumor suppressor p53 (p53, 1:1000, cat. no. 2524, Cell Signaling Technology), phosphorylated at Ser15 p53 (p-p53, 1:1000, cat. no. 9284, Cell Signaling Technology), C/EBP homologous protein (CHOP; 1:1000; cat. no. 5554, Cell Signaling Technology), Bcl-2-associated X protein (Bax; 1:1000; cat. no. 5023, Cell Signaling Technology), caspase-9 (C9, 1:100, cat. no. sc-133109, Santa Cruz Biotechnology), DR5 death receptor 5 (DR5, 1:500, cat. no. CSB-PA018500, Cusabio Technology), caspase-8 (C8, 1:200, cat. no. sc-5263, Santa Cruz Biotechnology), activated cleaved caspase-3 (CC3, 1:1000, cat. no. ab13847, Abcam), p21 Waf1/Cip1 (p21, 1:1000, cat. no. 37543, Cell Signaling Technology), p16 INK4A (p16, 1:1000, cat. no. 80772, Cell Signaling Technology), marker of proliferation Ki-67 (Ki-67, 1:1000, cat no. NBP2-22112, Novus Biologicals), β-galactosidase (GLB-1, 1:500, cat. no. ab55176, Abcam), α-smooth muscle actin (α-SMA, 1:100, cat no. sc-130617; Santa Cruz Biotechnology), and β-actin (1:2500, cat. no. 4967, Cell Signaling Technology). As secondary antibodies, the anti-rabbit IgG, HRP-linked antibody (1:1000, cat. no. 7074, Cell Signaling Technology) or the anti-mouse IgG, HRP-linked antibody (1:1000, cat. no. 7076, Cell Signaling Technology) were used. All western blot results are provided as Appendix A of the manuscript.

### 4.3. Detection of Cell Necrosis, ROS Production, IL-1β, and TGF-β1

Cell necrosis due to exposure to high albumin concentration and the toxicity of the used compounds was assessed colorimetrically with lactate dehydrogenase (LDH) release assay in RPTECs cultured in 96-well plates. The Cytotox Non-Radioactive Cytotoxic Assay kit (Promega Corporation, Madison, WI, USA) was used, and cytotoxicity was calculated using the equation Cytotoxicity (%) = (LDH in the supernatant: Total LDH) × 100. These experiments were repeated three times.

Reactive oxygen species (ROS) production was evaluated in RPTECs cultured in 96-well plates. Following a 24-h period of cell culture, RPTECs were treated with 5 μM of the fluorogenic probe CellROX Deep Red Reagent (Invitrogen, Life Technologies, Carlsbad, CA, USA) for 30 min at 37 °C. Subsequently, the RPTECs were rinsed with phosphate-buffered saline (Sigma-Aldrich; Merck Millipore), and the fluorescence signal intensity was measured using an EnSpire Multimode Plate Reader (PerkinElmer, Waltham, MA, USA). These experiments were conducted in triplicate.

Interleukin-1β (IL-1β) and transforming growth factor-β1 (TGF-β1) were measured in the supernatant of RPTECs cultured in 6-well plates. For IL-1β, the Hunan IL-1β PLATINUM ELISA kit (ca. no. BMS224/2, Bender MedSystems, Vienna, Austria) with a sensitivity of 0.3 pg/mL was used. The concentration of TGF-β1 was quantified using the Human TGF-beta-1 ELISA Kit (AssayPro, St. Charles, MO, USA). This kit has a detection range of 31–2000 pg/mL. ELISA measurements were conducted using the EnSpire Multimode Plate Reader (Perkin Elmer). The experiments were replicated three times.

### 4.4. Statistical Analysis

Statistical analysis was performed using IBM SPSS Statistics for Windows, Version 26 (IBM Corp., Armonk, NY, USA). The one-way analysis of variance (ANOVA) was used to compare means. Error bars represent the standard error of means, and *p*-values less than 0.05 were considered statistically significant. The GraphPad Prism, Version 9 (GraphPad Software, Boston, MA, USA) was used for the graphical presentation of the results.

## Figures and Tables

**Figure 1 ijms-24-09640-f001:**
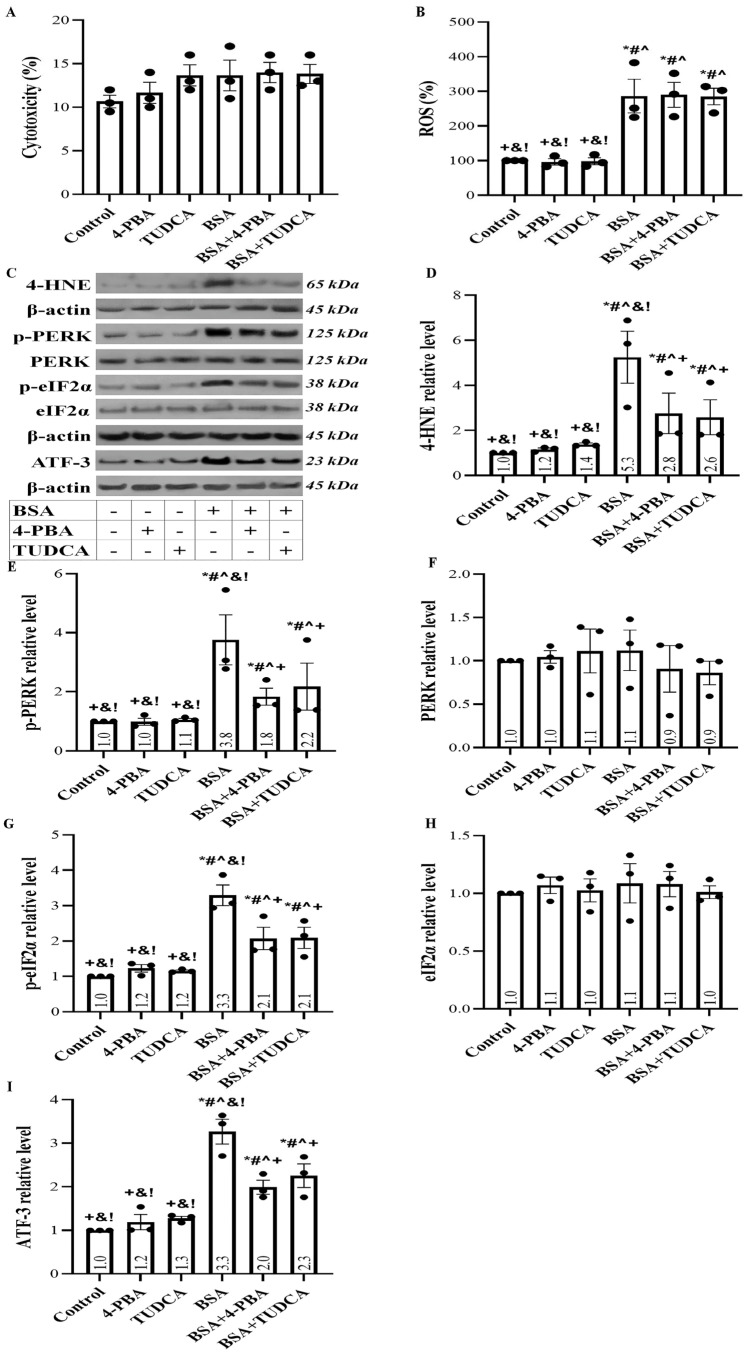
Albumin overload does not induce cell necrosis but triggers ROS production and UPR. LDH release assay showed that albumin overload does not cause cell necrosis. TUDCA and 4-PBA were not cytotoxic (**A**). High albumin concentration induced a ROS burst in RPTECs. Neither TUDCA nor 4-PBA affected ROS production (**B**). Panel (**C**) depicts the results of one out of three performed experiments. BSA treatment increased the level of 4-HNE-modified proteins. TUDCA and 4-PBA partially ameliorated protein modification (**D**). In RPTECs, high albumin concentration triggered a UPR since the p-PERK level increased (**E**), whereas the total PERK remained unaffected (**F**). p-eIF2α level also enhanced (**G**), whereas the total eIF2α remained stable (**H**). ATF3 level upregulated (**I**). TUDCA and 4-PBA ameliorated but did not eliminate all the above changes. * *p* < 0.05 vs. control; # *p* < 0.05 vs. RPTECs treated with 4-PBA; ^ *p* < 0.05 vs. RPTECs treated with TUDCA; + *p* < 0.05 vs. RPTECs exposed to BSA; & *p* < 0.05 vs. RPTECs exposed to BSA and 4-PBA; ! *p* < 0.05 vs. RPTECs exposed to BSA and TUDCA. 4-PBA, 4-Phenylbutyric acid; 4-HNE, 4-Hydroxynonenal; ATF-3, activating transcription factor-3; BSA, bovine serum albumin; eIF2α, eukaryotic translation initiation factor-2α; PERK, PKR-like ER kinase; ROS, reactive oxygen species; TUDCA, tauroursodeoxycholic acid.

**Figure 2 ijms-24-09640-f002:**
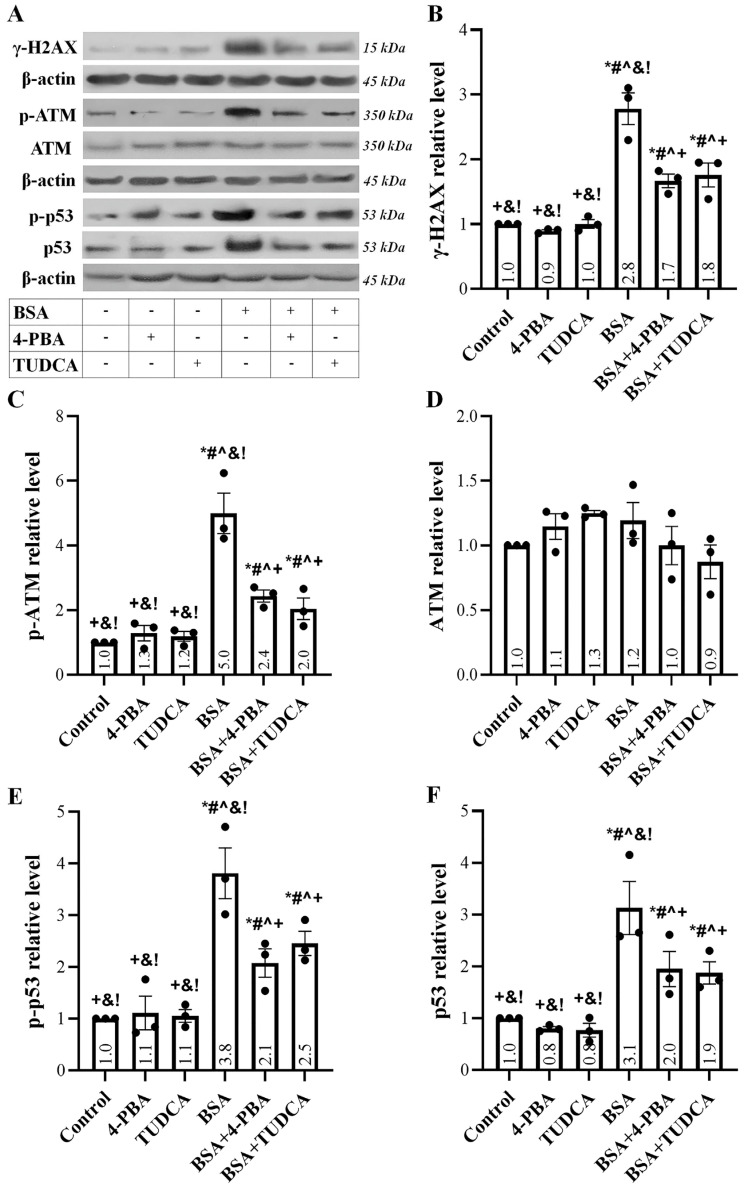
Albumin overload induces a DDR. Panel (**A**) depicts the results of one out of three performed experiments. Albumin overload caused DNA damage detected by the enhanced γ-H2AX level. TUDCA and 4-PBA decreased to some extent DNA damage (**B**). DNA damage in RPTECs exposed to albumin elicited a DDR since the p-ATM level increased (**C**) on a stable total ATM background (**D**). As a result of ATM activation, p-p53 increased (**E**), resulting in total p53 upregulation (**F**). TUDCA and 4-PBA reduced to some extent all the above albumin overload-induced changes. * *p* < 0.05 vs. control; # *p* < 0.05 vs. RPTECs treated with 4-PBA; ^ *p* < 0.05 vs. RPTECs treated with TUDCA; + *p* < 0.05 vs. RPTECs exposed to BSA; & *p* < 0.05 vs. RPTECs exposed to BSA and 4-PBA; ! *p* < 0.05 vs. RPTECs exposed to BSA and TUDCA. 4-PBA, 4-Phenylbutyric acid; γ-H2AX, phosphorylated at Ser139 histone H2AX; ATM, ataxia telangiectasia mutated kinase; BSA, bovine serum albumin; p53, tumor suppressor p53; TUDCA, tauroursodeoxycholic acid.

**Figure 3 ijms-24-09640-f003:**
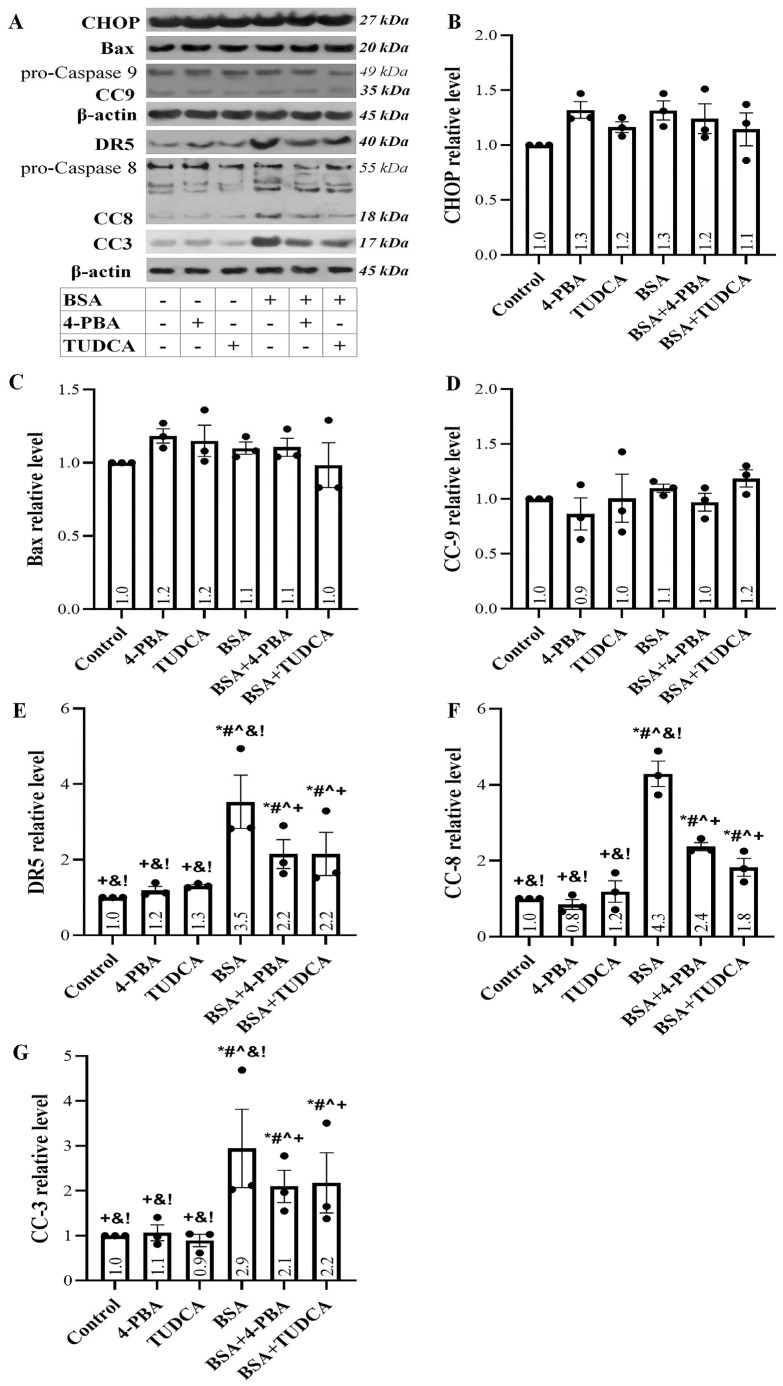
Albumin overload triggers the extrinsic but not the intrinsic apoptotic pathway. Panel (**A**) depicts the results of one out of three performed experiments. Albumin overload left the intrinsic apoptotic pathway unaffected since the CHOP (**B**), Bax (**C**), and CC-9 (**D**) levels did not change significantly. TUDCA and 4-PBA did not change the levels of the above proteins. Exposure of the RPTECs to high albumin concentration triggered the extrinsic apoptotic pathway as it upregulated DR5 (**E**) and CC-8 (**F**). Eventually, apoptosis ensued since the level of CC-3 increased (**G**). TUDCA and 4-PBA ameliorated but did not eliminate the above changes. * *p* < 0.05 vs. control; # *p* < 0.05 vs. RPTECs treated with 4-PBA; ^ *p* < 0.05 vs. RPTECs treated with TUDCA; + *p* < 0.05 vs. RPTECs exposed to BSA; & *p* < 0.05 vs. RPTECs exposed to BSA and 4-PBA; ! *p* < 0.05 vs. RPTECs exposed to BSA and TUDCA. 4-PBA, 4-Phenylbutyric acid; BSA, bovine serum albumin; Bax, Bcl-2-associated X protein; CC-3, cleaved caspase-3; CC-8, cleaved caspase-8; CC-9, cleaved caspase-9; CHOP, C/EBP homologous protein; DR5, death receptor-5; TUDCA, tauroursodeoxycholic acid.

**Figure 4 ijms-24-09640-f004:**
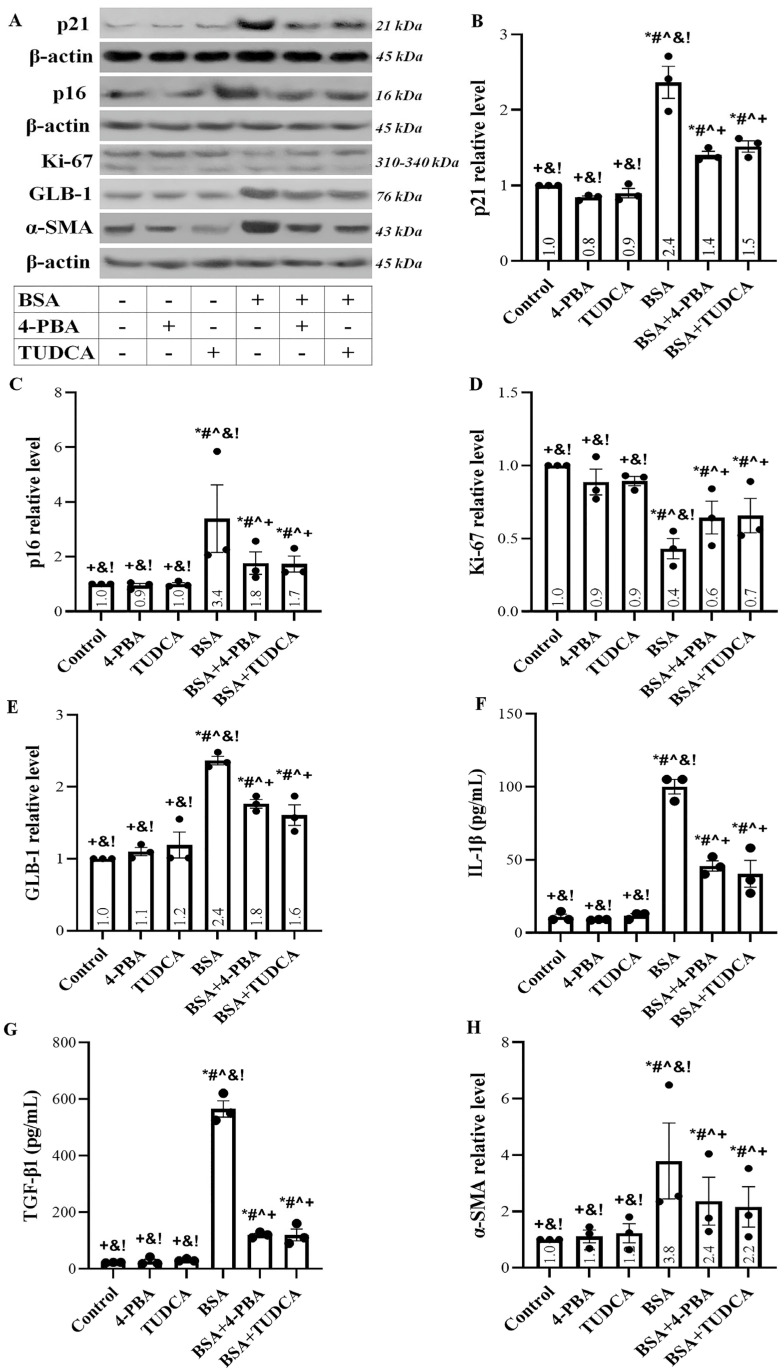
Albumin overload induces cellular senescence and EMT. Panel (**A**) depicts the results of one out of three performed experiments. Albumin overload increased the expression of the cell cycle arrest inducers p21 (**B**) and p16 (**C**). In addition, it decreased the level of the cell proliferation marker Ki-67 (**D**) and enhanced the cellular senescence marker GLB-1 (**E**). RPTECs exposed to high albumin concentration overproduced IL-1β (**F**) and TGF-β1 (**G**). Exposure of RPTECs to high albumin concentration resulted in EMT as the level of α-SMA increased (**H**). TUDCA and 4-PBA ameliorated but did not eliminate all the changes above. * *p* < 0.05 vs. control; # *p* < 0.05 vs. RPTECs treated with 4-PBA; ^ *p* < 0.05 vs. RPTECs treated with TUDCA; + *p* < 0.05 vs. RPTECs exposed to BSA; & *p* < 0.05 vs. RPTECs exposed to BSA and 4-PBA; ! *p* < 0.05 vs. RPTECs exposed to BSA and TUDCA. 4-PBA, 4-Phenylbutyric acid; α-smooth muscle actin; BSA, bovine serum albumin; GLB-1, β-galactosidase; IL-1β, interleukin-1β; Ki-67, marker of proliferation Ki-67; p16, p16 INK4A; p21, p21 Waf1/Cip1; TGF-β1, transforming growth factor-β1; TUDCA, tauroursodeoxycholic acid.

**Figure 5 ijms-24-09640-f005:**
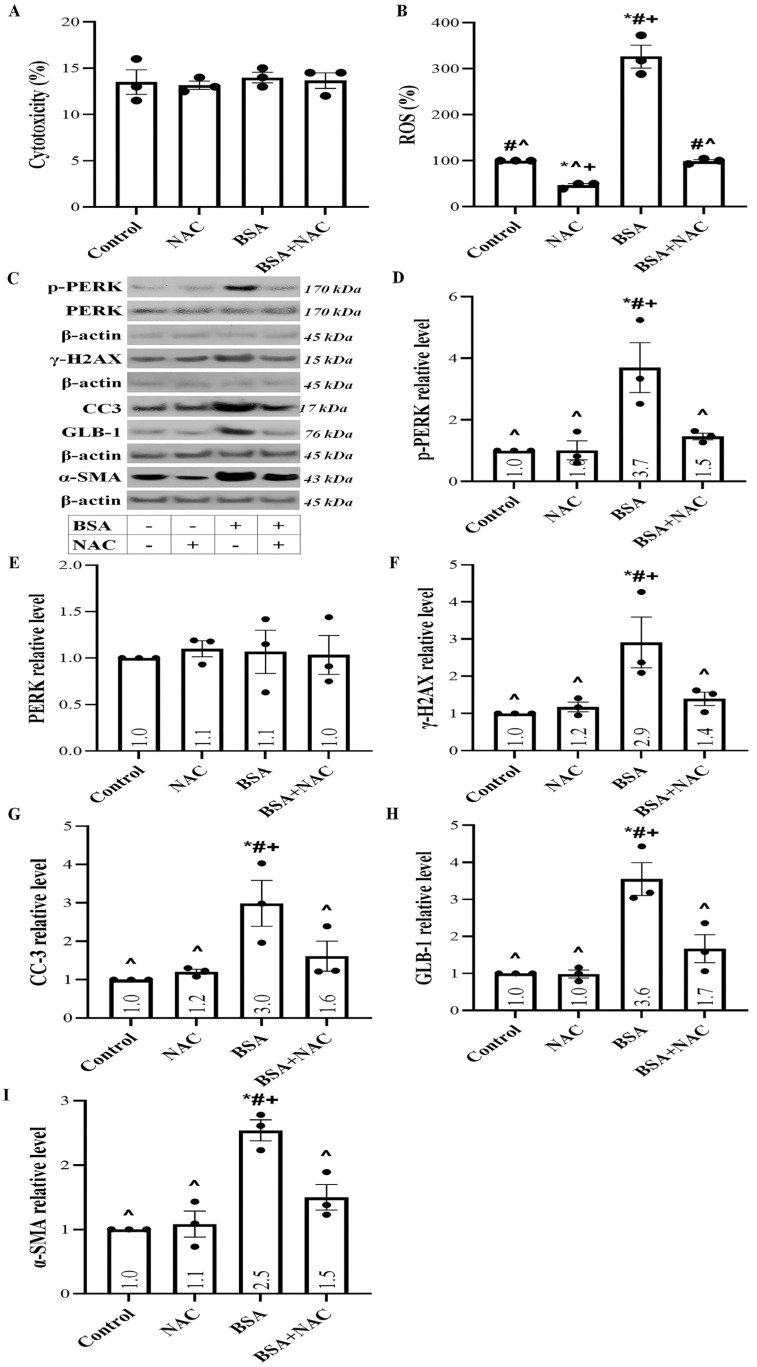
NAC at a non-cytotoxic concentration prevents ROS upregulation, initiation of UPR and DDR, and eventually apoptosis, senescence, and EMT. LDH release assay revealed that NAC was not cytotoxic for the RPTECs at the used concentration (**A**). NAC completely abolished the rise of ROS levels in RPTECs exposed to high albumin concentration (**B**). Panel (**C**) depicts the results of one out of three performed experiments. NAC prevented UPR since it normalized the albumin overload-induced increase in the p-PERK level (**D**) without affecting total PERK (**E**). NAC prevented DDR as it normalized the albumin overload-induced increase in the γ-H2AX level (**F**). NAC prevented the harmful effects of apoptosis, senescence, and EMT. NAC normalized the albumin overload-induced upregulation of CC-3 (**G**), GLB-1 (**H**), and α-SMA (**I**). * *p* < 0.05 vs. control; # *p* < 0.05 vs. RPTECs treated with NAC; ^ *p* < 0.05 vs. RPTECs exposed to BSA; + *p* < 0.05 vs. RPTECs exposed to BSA and NAC. α-SMA, α-smooth muscle actin; γ-H2AX, phosphorylated at Ser139 histone H2AX; BSA, bovine serum albumin; CC-3, cleaved caspase-3; GLB-1, β-galactosidase; NAC, N-acetylcysteine; PERK, PKR-like ER kinase; ROS, reactive oxygen species.

**Figure 6 ijms-24-09640-f006:**
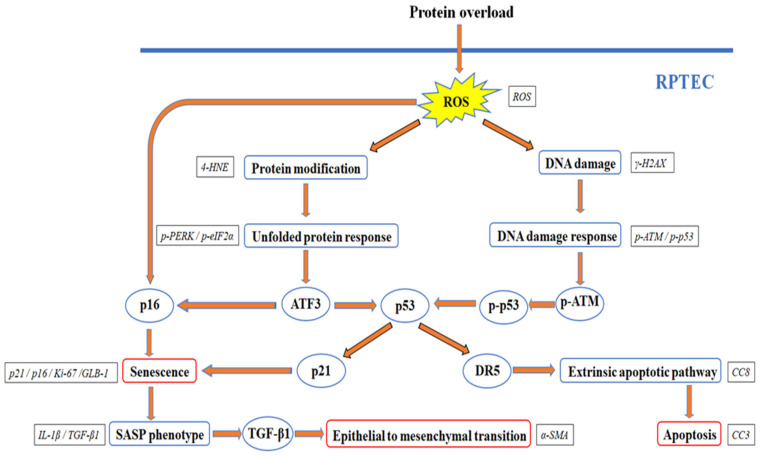
In RPTECs, albumin overload triggers a UPR and DDR, resulting in apoptosis, senescence, and EMT. Albumin overload causes ROS overproduction. ROS induces protein modification triggering a UPR. In addition, ROS causes DNA damage eliciting a DDR. Both DDR and UPR result in p53 upregulation, which induces apoptosis through the extrinsic apoptotic pathway. Both DDR and UPR, through p53, upregulate p21, while p16 increases due to the UPR-dependent ATF-3 or ROS. Upregulation of the above cell cycle arrest inducers causes cellular senescence and a SASP. TGF-β1 overproduction because of the SASP may contribute to the albumin overload-induced EMT of the RPTECs. 4-HNE, 4-Hydroxynonenal; α-SMA, α-smooth muscle actin; γ-H2AX, phosphorylated at Ser139 histone H2AX; ATF-3, activating transcription factor-3; ATM, ataxia telangiectasia mutated kinase; CC-3, cleaved caspase-3; CC-8, cleaved caspase-8;, CC-9, cleaved caspase-9; DR5, death recetror-5; eIF2α, eukaryotic translation initiation factor-2α; GLB-1, β-galactosidase; IL-1β, interleukin-1β; Ki-67, marker of proliferation Ki-67; p16, p16 INK4A; p21, p21 Waf1/Cip1; p53, tumor suppressor p53; PERK, PKR-like ER kinase; ROS, reactive oxygen species; RPTEC, renal proximal tubular epithelial cell; SASP, senescence-associated secretory phenotype; TGF-β1, transforming growth factor-β1.

## Data Availability

The analyzed datasets generated during the study are available from the corresponding author upon reasonable request.

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
