# Peer review of "Routes of Albumin Overload Toxicity in Renal Tubular Epithelial Cells"

_ijms, 2023, doi:10.3390/ijms24119640_

Round 1

Reviewer 1 Report

This work shows the routes of albumin toxicity on renal epithelial cells and evaluates ERS blockers versus NAC mediated difference in protecting the epithelium. The authors may address the following concerns before publication of work:

The authors should use multiple concentrations of blockers/albumin to conclusively establish the claim either in favor or against ERS blockers as the authors themselves mentioned the usage of high but not toxic concentration of NAC alone. The authors should provide justification for usage of single dose of blockers/albumin and state that in limitations of the study. The authors may use individual data points within the bar graphs to improve the clarity in results/figures. Also, the statistical markers and font size in figures are too small, and hence the data may need better presentation. The limitations mentioned regarding other confounding factors may not be relevant and apply to most of the in vitro studies. It would be interesting to evaluate the markers studied here in clinical samples in case they are available to the authors. 

Minor English language editing may be necessary.

Author Response

First, we would like to thank the reviewer since his/her valuable comments helped us improve our manuscript’s quality.

  1. The authors should use multiple concentrations of blockers/albumin to conclusively establish the claim either in favor or against ERS blockers as the authors themselves mentioned the usage of high but not toxic concentration of NAC alone. The authors should provide justification for usage of single dose of blockers/albumin and state that in limitations of the study.

Thank you for allowing us to clarify this point.

“The BSA, 4-PBA, TUDCA, and NAC concentrations have been selected after preliminary experiments with concentrations within the range used in previous studies [10,26,29,52-54]. RPTECs were cultured with 10 or 30 mg/mL BSA, and p-PERK was used as the outcome. We selected the concentration with the highest effect. For 4-PBA and TUDCA, we cultured RPTECs with 1 or 2 mM of these compounds. Again p-PERK was the outcome. We selected the concentration with the highest effect. We used 1 or 3 mg/mL of NAC, and ROS production was the outcome. Again, we selected the concentration with the highest effect. The LDH release assay revealed that the selected concentrations were not cytotoxic for the RPTECs.”

 We incorporated this information in the methods section of the revised manuscript.

  1. The authors may use individual data points within the bar graphs to improve the clarity in results/figures. Also, the statistical markers and font size in figures are too small, and hence the data may need better presentation.

Thank you for the comment. We revised five figures in the manuscript according to the recommendation.

  1. The limitations mentioned regarding other confounding factors may not be relevant and apply to most of the in vitro studies. It would be interesting to evaluate the markers studied here in clinical samples in case they are available to the authors.

Unfortunately, at present, we do not have readily available clinical samples; however, following the suggestion, we will arrange a co-elaboration with the Pathology Department of our University.

In the revised manuscript, we made a notion in the limitations paragraph.

“Certainly, the simultaneous assessment of markers associated with DDR, ERS, apoptosis, senescence, and EMT in clinical samples obtained from patients with albuminuria would be highly intriguing.”

Reviewer 2 Report

In this work “Routes of albumin overload toxicity in renal tubular epithelial cells”, the authors propose a work where they evaluated whether an unfolded protein response (UPR) or DNA damage response (DDR) is elicited in RPTECs exposed to high albumin concentration, evaluating the deleterious outcomes of the above pathways, apoptosis, senescence, or epithelial-to-mesenchymal transition (EMT). 

The purpose of this work should be highlighted more, going to highlight the implications and therapeutic applicability.

The choice of the two molecules (TUDCA and 4-PBA) should be justified and framed in the context of the pathology.

-The work is enough complete, but is enough lacking on molecular pathway description on unfolded protein response (UPR) or DNA damage response (DDR).

-Albumin overload triggers UPR and DDR and induces apoptosis, senescence, or EMT by increasing ROS: in this section it is not clear why they detected interleukin-1β (IL-1β) and transforming growth factor-β1 (TGF-β1). They are not specific markers. Why did not use MTT and β-gal assay in the experiments? Please justify this choice.

- Why it was specifically chosen (TUDCA) and 4-Phenylbutyric acid (4-PBA) for the experiments? Also insert the scientific references for the choice of these two drugs.

- How can you be sure you have worked with RPTEC cells?

- With what criterion were the quantities of the molecules chosen for TUDCA and 4-PBA ?

The English language requires minimal editing and blends well with the context of the paper.

Author Response

First, we would like to thank the reviewer since his/her encouraging comments helped us to improve our manuscript.

1. The purpose of this work should be highlighted more, going to highlight the implications and therapeutic applicability.

The purpose and the possible clinical implications of the study were emphasized to a greater extent in the introduction section of the revised manuscript.

“With the introduction of anti-ERS agents into clinical practice [12,13], it would be beneficial to elucidate the significance of this pathway, as well as the role of DDR, in the toxicity of albumin overload in RPTECs. This clarification would provide a more accurate understanding of the potential impact of anti-ERS agents on patients with albuminuria.”

2. The choice of the two molecules (TUDCA and 4-PBA) should be justified and framed in the context of the pathology.

In the discussion section of the revised manuscript, with references to experimental studies and the clinical implications of these compounds, we justified their selection as anti-ERS agents in our study.

“Both compounds inhibit ERS due to their chaperoning activity [24,25], and experimental studies confirmed that they protect RPTECs from ERS [26-29]. The fact that these substances are already in clinical use, 4-PBA to treat congenital diseases in the urea cycle [30], and TUDCA for treating primary biliary cholangitis [31], make them particularly appealing for further investigation since we already know much about their pharmacologic properties and side-effects. Currently, both medications are under investigation as anti-ERS agents for the treatment of various diseases [24,30], and recently a combination of them has been approved for the treatment of amyotrophic lateral sclerosis [13].”

3. The work is enough complete, but is enough lacking on molecular pathway description on unfolded protein response (UPR) or DNA damage response (DDR). 

Since too many molecules are implicated in these pathways, we choose to evaluate key points of those pathways. For instance, from the three UPS pathways, we chose to assess the PERK pathway. The activation of the latter confirms that a UPR takes place.

4. Albumin overload triggers UPR and DDR and induces apoptosis, senescence, or EMT by increasing ROS: in this section it is not clear why they detected interleukin-1β (IL-1β) and transforming growth factor-β1 (TGF-β1). They are not specific markers. Why did not use MTT and β-gal assay in the experiments? Please justify this choice.

We agree that IL-1β and TGF-β1 are not specific markers of senescence. The reason for detecting these cytokines is to confirm that the senescent cells also acquired a SASP phenotype. In the discussion, we wrote about it:
“In addition, RPTECs acquired a senescence-associated secretory phenotype as they overproduced IL-1β and TGF-β1.”

Generally, we avoid using MTT or XTT assay for detecting cell proliferation, since these assays are based on mitochondrial function and especially in NADH production.
However, mitochondrial function under various stress conditions altered irrespectively of the cell proliferation state. In our lab, we prefer BrdU assay or the proliferation marker Ki-67. We selected the latter because maybe not too much BrdU incorporation in the DNA of new cells takes place within 24 hours of the experiment. In the revised manuscript, we added two references showing the value of Ki-67 as a proliferation marker. Regarding the β-Gal assay, we straightforwardly assess the expression of this enzyme (GLB-1) with WB.

5. Why it was specifically chosen (TUDCA) and 4-Phenylbutyric acid (4-PBA) for the experiments? Also insert the scientific references for the choice of these two drugs.

“Both compounds inhibit ERS due to their chaperoning activity [24,25], and experimental studies confirmed that they protect RPTECs from ERS [26-29]. The fact that these substances are already in clinical use, 4-PBA to treat congenital diseases in the urea cycle [30], and TUDCA for treating primary biliary cholangitis [31], make them particularly appealing for further investigation since we already know much about their pharmacologic properties and side-effects. Currently, both medications are under investigation as anti-ERS agents for the treatment of various diseases [24,30], and recently a combination of them has been approved for the treatment of amyotrophic lateral sclerosis [13].”

6. How can you be sure you have worked with RPTEC cells?

We purchased the cells from ScienCell, Carlsbad, CA, USA. According to the company, these cells above are differentiated, well‑characterized passage one RPTECs. We noted this in the methods subsection.

7. With what criterion were the quantities of the molecules chosen for TUDCA and 4-PBA ?

We clarify this in the methods subsection of the revised manuscript. “The BSA, 4-PBA, TUDCA, and NAC concentrations have been selected after preliminary experiments with concentrations within the range used in previous studies [10,26,29,53-55]. RPTECs were cultured with 10 or 30 mg/mL BSA, and p-PERK was used as the outcome. We selected the latter concentration, which had the highest effect. For 4-PBA and TUDCA, RPTECs cultured with 30 mg/mL BSA were exposed 1 or 2 mM of these compounds. Again p-PERK was the outcome. We selected the concentration with the highest effect. We used 1 or 3 mg/mL of NAC, and ROS production was the outcome. Again, we selected the concentration with the highest effect. The LDH release assay revealed that the selected concentrations were not cytotoxic for the RPTECs.”